# Numerical Solution of the Electric Field and Dielectrophoresis Force of Electrostatic Traveling Wave System

**DOI:** 10.3390/mi14071347

**Published:** 2023-06-30

**Authors:** Yue Yu, Yao Luo, Jan Cilliers, Kathryn Hadler, Stanley Starr, Yanghua Wang

**Affiliations:** 1Resource Geophysics Academy, Imperial College London, London SW7 2BP, UK; yy2120@ic.ac.uk (Y.Y.); kathryn.hadler@list.lu (K.H.); yanghua.wang@imperial.ac.uk (Y.W.); 2Department of Earth Science and Engineering, Imperial College London, London SW7 2AZ, UK; s.starr@imperial.ac.uk; 3School of Electrical Engineering and Automation, Wuhan University, Wuhan 430072, China

**Keywords:** boundary element method, cell manipulation and separation, charge simulation method, dielectrophoretic force, parallel electrodes, electric field calculation, electrostatic traveling wave

## Abstract

Electrostatic traveling wave (ETW) methods have shown promising performance in dust mitigation of solar panels, particle transport and separation in in situ space resource utilization, cell manipulation, and separation in biology. The ETW field distribution is required to analyze the forces applied to particles and to evaluate ETW design parameters. This study presents the numerical results of the ETW field distribution generated by a parallel electrode array using both the charge simulation method (CSM) and the boundary element method (BEM). A low accumulated error of the CSM is achieved by properly arranging the positions and numbers of contour points and fictitious charges. The BEM can avoid the inconvenience of the charge position required in the CSM. The numerical results show extremely close agreement between the CSM and BEM. For simplification, the method of images is introduced in the implementation of the CSM and BEM. Moreover, analytical formulas are obtained for the integral of Green’s function along boundary elements. For further validation, the results are cross-checked using the finite element method (FEM). It is found that discrepancies occur at the ends of the electrode array. Finally, analyses are provided of the electric field and dielectrophoretic (DEP) components. Emphasis is given to the regions close to the electrode surfaces. These results provide guidance for the fabrication of ETW systems for various applications.

## 1. Introduction

An electrostatic traveling wave (ETW) field can be produced by a set of electrodes, insulated from each other and connected to AC poly-phase voltage sources. Either neutral or charged fine particles brought into such a field move due to the action of electric forces, gravitational forces, and other forces related to their different physical properties. The ETW method originates from dielectrophoresis (DEP), first defined by Pohl as the interaction between non-uniform electric fields with polarized particles and liquids [1,2]. The researchers who followed have developed a generalized theory for calculating DEP and explored its various practical applications, such as the precipitation and dispersion of liquid [3,4]. Recently, this electrokinetic phenomenon has been applied in many diverse areas, such as xerographic particle transport in electrophotography, dust mitigation of solar panels, cell manipulation and separation in biology, and lunar particle transport and separation [5,6,7,8,9,10].

A theoretical understanding of particle movement in the ETW conveyer system will allow design and operating parameter evaluation. These particles are subject to many forces, including the Coulomb force, dielectrophoretic (DEP) force, gravitational force, friction, image force, and possibly fluid drag. Of these, the Coulomb and DEP forces are predominant, and their analysis requires the accurate calculation of the electric field [11,12]. The Accurate prediction of the ETW electric field distribution is, therefore, essential. 

### 1.1. Background of the Electric Potential Problem

The ETW field is generally produced by an array of parallel electrodes with the same width and thickness, insulated from each other, and connected to an AC, poly-phase voltage source. Figure 1 shows a diagram of a typical electrode system for particle transport and separation. A four-phase rectangular wave voltage source is often used in ETW systems [13,14,15], and is used here as an example in the calculations. 

The electrode length is much larger than both the electrode width and the gap between the electrodes, and the electric field model can be simplified to a 2D problem in the *x*–*y* plane, as shown in Figure 2. In phases 2 and 4, Dirichlet conditions *ϕ* = 0 can be designated on *x* = 0. In phases 1 and 3, the symmetry leads to the Neumann condition: ∂ϕ/∂n=0. Thus, the 2D electrostatic problem can be described by Laplace’s equation Δϕ=0, associated with the boundary conditions on the electrode surfaces, which vary in time according to the system function. 

### 1.2. Methods Development

Several approximate analytical and numerical methods have been proposed for the solution of the electric field produced by this particular arrangement of electrodes. In 1996, Wang and co-workers [16] used Green’s theorem to calculate the electric field for 2D electrode arrays. Morgan and co-workers [17,18] developed Fourier series methods and the finite element method (FEM) for calculating DEP and traveling wave forces generated by interdigitated electrode arrays. Sun and co-workers [19] completed an analytical solution using the Schwarz–Christoffel mapping method without any approximation of the boundary conditions. Gauthier and co-workers [20] developed a Fourier series method to calculate DEP force with two facing electrode arrays. However, in all these papers, either or both of the following two approximations have been applied: (1) the electrode boundaries are set as a line without considering the electrode shape, and (2) a linear potential variation between electrodes is assumed. These approximations deviate from reality in certain conditions and restrict the application scope of their methods.

In this paper, we present two alternative approaches, the charge simulation method (CSM) and the boundary element method (BEM), to calculate the electric field distribution in such an electrode arrangement. A similar approach to BEM, a boundary integral solution of a potential problem, has been obtained and has high computation efficiency using the Nyström method [21]. The CSM and BEM methods have the benefit of no requirement for any approximation of the boundary conditions or the electrode shape and can accurately predict the electric field on the electrode edges and gaps efficiently. The formulas we derived can be adapted to various designs and parameters easily.

## 2. Theory of the Charge Simulation Method (CSM)

### 2.1. Basic Principle

The charge simulation method is based on the concept of discrete charges, which has proven to be very powerful and efficient for solving many electrostatic problems [22]. Masuda and Kamimura [23] used a similar substitute charge method to calculate the electric field of parallel cylindrical electrodes.

The CSM is a numerical method for the application of the Trefftz method for the solution of the boundary value problem (BVP) in an electrostatic field where the partial differential equation is satisfied perfectly while the boundary conditions are satisfied approximately [24,25].

The basis of the CSM is the use of a group of discrete, fictitious charges as particular solutions of Poisson equations, where the distributed charges on the boundary are replaced by these discrete charges arranged outside the boundary. The fictitious charges are shown as the hollow circles in the first quadrant inside the boundary, and (*x*_f_^(i)^, *y*_f_^(i)^) represents the position of the i-th fictitious charge. Using the method of images, the image fictitious charges are applied in the other three quadrants, and the positions can be easily obtained according to the symmetries. In the CSM, the solution of Laplace’s equation depends on the determination of the values of these fictitious charges. The arrangement of fictitious charges and contour points is illustrated in Figure 3. The contour points are on the boundary of electrodes with known potentials, which can be used in the equations to solve the unknown fictitious charges. The contour points are shown as the symbol × on the boundary, and (*x*_e_^(i)^, *y*_e_^(i)^) represents the position of the i-th contour point. Γ*_e_* and Γ*_f_* represent the *e*-th and *f*-th boundaries, while Γ*_e_*^′^ and Γ*_f_*^′^ are the image boundaries, respectively. *r*_1_, *r*_2_, *r*_3_, and *r*_4_ are the distances between the contour point and the fictitious charge and image fictitious charges. *d*_1_, *s*_1_, *s*_2,_ and *δ* are simulation parameters that are used to determine the positions of contour points and fictitious charges, which can be referred to in the following calculations.

The potential resulting from the superposition of discrete charges must be equal to the boundary potential *ϕ*_C_ on the electrode surfaces:(1)$$\phi_{f}=\sum_{e=1}^{w}\left(\sum_{i=1}^{n}P_{e,f}^{\left(i,j\right)}Q_{f}^{j}\right)$$
where $P_{e,f}^{\left(i,j\right)}$ is the associated potential coefficient that can be found from the fundamental solution of Poisson’s equation, and it only depends on the related position between the calculating point and the *n*-th fictitious charge.
(2)$$P_{e,f}^{\left(i,j\right)}=-\frac{1}{2\pi \varepsilon_{0}}\cdot \ln\sqrt{{\left(x_{e}^{\left(i\right)}-x_{f}^{\left(j\right)}\right)}^{2}+{\left(y_{e}^{\left(i\right)}-y_{f}^{\left(j\right)}\right)}^{2}}$$

$Q_{f}^{j}$ is the value of the *j*-th fictitious charge on the *f*-th electrode. *n* is the total number of fictitious charges in an electrode. *w* is the total number of electrodes. Since there is a unique solution to this boundary value problem, the potential can be found unambiguously from Equation (1), where *ε*_0_ is the permittivity of air and ($x_{e}^{\left(i\right)}$, $y_{e}^{\left(i\right)}$) and ($x_{f}^{\left(j\right)}$, $y_{f}^{\left(j\right)}$) represent the positions of the *i*-th fictitious charge of the *e*-th electrode and the *j*-th points on the surface of the *f*-th electrodes.

The application of Equations (1) and (2) leads to a system for *w* electrodes, each with *N* discrete charges:(3)$$\left[\begin{array}{llll} P_{11} & P_{12} & \ldots & P_{1w}\\ P_{21} & P_{22} & \ldots & P_{2w}\\ \vdots & \vdots & \ddots & \vdots \\ P_{w1} & P_{w2} & \ldots & P_{ww} \end{array}\right]\;\left[\begin{array}{l} Q_{1}\\ Q_{2}\\ \;\vdots \\ Q_{w} \end{array}\right]=\;\left[\begin{array}{l} \phi_{1}\\ \phi_{2}\\ \;\vdots \\ \phi_{w} \end{array}\right]$$
where the elements of the submatrices are
(4)$$P_{ef}=\left[\begin{array}{llll} P_{e,f}^{\left(1,1\right)} & P_{e,f}^{\left(1,2\right)} & \ldots & P_{e,f}^{\left(1,N\right)}\\ P_{e,f}^{\left(2,1\right)} & P_{e,f}^{\left(2,2\right)} & \ldots & P_{e,f}^{\left(2,N\right)}\\ \vdots & \vdots & \ddots & \vdots \\ P_{e,f}^{\left(N,1\right)} & P_{e,f}^{\left(N,2\right)} & \ldots & P_{e,f}^{\left(N,N\right)} \end{array}\right]$$
(5)$$Q_{f}={\left[\begin{array}{ccc} \begin{array}{ccc} \begin{array}{cc} q_{f}^{\left(1\right)} & q_{f}^{\left(2\right)} \end{array} & . & . \end{array} & . & q_{f}^{\left(N\right)} \end{array}\right]}^{T}$$
(6)$$\phi_{f}={\left[\begin{array}{ccc} \begin{array}{ccc} \begin{array}{cc} \phi_{f} & \phi_{f} \end{array} & . & . \end{array} & . & \phi_{f} \end{array}\right]}^{T}$$
where $q_{f}^{\left(j\right)}\;$ is the value of the fictitious charge of ($x_{f}^{\left(j\right)}$, $y_{f}^{\left(j\right)}$). $\phi_{f}$ is the potential of the *f*-th electrode. Equation (3) can be expressed in a simplified form:(7)P Q=ϕ
where **P** is the potential coefficient matrix, **Q** is the column vector of unknown charges, and ϕ is the potential of the points on the electrode boundary. 

### 2.2. Implementation of CSM

Considering the symmetry of the electrode array and the applied voltage, the method of images can be used to simplify the analysis. Based on the center position of *o* in Figure 3, the boundaries are evenly distributed in the four quadrants of the coordinate, and the fictitious charges in the first quadrant are used as the origin charges. The combination of the associated potential coefficient of the fictitious line charges and its image fictitious line charges in the other three quadrants (see Figure 3) can be expressed as follows:(8)$$P_{e,f}^{\left(i,j\right)*}=\left\{\begin{array}{l} \frac{1}{2\pi \varepsilon_{0}}\ln\frac{r_{2}r_{3}}{r_{1}r_{4}}\;\;\mathrm{for}\;\mathrm{phase}\;2\;\mathrm{and}\;4\\ \frac{1}{2\pi \varepsilon_{0}}\ln\frac{1}{r_{1}r_{2}r_{3}r_{4}}\;\;\mathrm{for}\;\mathrm{phase}\;1\;\mathrm{and}\;3 \end{array}\right.$$
where
(9a)$$r_{1}=\sqrt{{\left(x_{e}^{\left(i\right)}-x_{f}^{\left(j\right)}\right)}^{2}+{\left(y_{e}^{\left(i\right)}-y_{f}^{\left(j\right)}\right)}^{2}}$$
(9b)$$r_{2}=\sqrt{{\left(x_{e}^{\left(i\right)}+x_{f}^{\left(j\right)}\right)}^{2}+{\left(y_{e}^{\left(i\right)}-y_{f}^{\left(j\right)}\right)}^{2}}$$
(9c)$$r_{3}=\sqrt{{\left(x_{e}^{\left(i\right)}+x_{f}^{\left(j\right)}\right)}^{2}+{\left(y_{e}^{\left(i\right)}+y_{f}^{\left(j\right)}\right)}^{2}}$$
(9d)$$r_{4}=\sqrt{{\left(x_{e}^{\left(i\right)}-x_{f}^{\left(j\right)}\right)}^{2}+{\left(y_{e}^{\left(i\right)}+y_{f}^{\left(j\right)}\right)}^{2}}$$

The two definitions of the associated potential coefficient in (8) are automatically satisfied with Dirichlet and Neumann conditions, respectively. Formula (2) is a generalized expression for each situation.

In this way, Equation (7) can be modified as follows:(10)$$\left[\begin{array}{llll} P_{11}{}^{*} & P_{12}{}^{*} & \ldots & P_{1,w/2}{}^{*}\;\\ P_{21}{}^{*} & P_{22}{}^{*} & \ldots & P_{2,w/2}{}^{*}\\ \vdots & \vdots & \ddots & \vdots \\ P_{w/2,1}{}^{*} & P_{w/2,2}{}^{*} & \ldots & P^{*}{}_{w/2,w/2} \end{array}\right]\;\left[\begin{array}{l} Q_{1}\\ Q_{2}\\ \;\vdots \\ Q_{w/2} \end{array}\right]=\;\left[\begin{array}{l} \phi_{1}\\ \phi_{2}\\ \;\vdots \\ \phi_{w/2} \end{array}\right]$$
where the submatrices are given by
(11)$$P_{ef}{}^{*}=\left[\begin{array}{llll} P_{e,f}^{\left(1,1\right)*} & P_{e,f}^{\left(1,2\right)*} & \ldots & P_{e,f}^{\left(1,N/2\right)*}\\ P_{e,f}^{\left(2,1\right)*} & P_{e,f}^{\left(2,2\right)*} & \ldots & P_{e,f}^{\left(2,N/2\right)*}\\ \vdots & \vdots & \ddots & \vdots \\ P_{e,f}^{\left(N/2,1\right)*} & P_{e,f}^{\left(N/2,2\right)*} & \ldots & P_{e,f}^{\left(N/2,N/2\right)*} \end{array}\right]$$

After obtaining the value of the fictitious charge in each position, the potential above the electrodes can be found with the following:(12)$$\phi \left(x,y\right)=P^{*}\left(x,y\right)Q$$
where
(13)$$P^{*}\left(x,y\right)=\left[\begin{array}{ccc} \begin{array}{ccc} \begin{array}{cc} P_{}^{\left(1\right)*}\left(x,y\right) & P_{}^{\left(2\right)*}\left(x,y\right) \end{array} & . & . \end{array} & . & P_{}^{\left(N/2\right)*}\left(x,y\right) \end{array}\right]$$
(14)$$Q={\left[\begin{array}{ccc} \begin{array}{ccc} \begin{array}{cc} q_{}^{\left(1\right)} & q_{}^{\left(2\right)} \end{array} & . & . \end{array} & . & q_{}^{\left(N/2\right)} \end{array}\right]}^{T}$$
and
(15)$$P_{}^{\left(n\right)*}\left(x,y\right)=\left\{\begin{array}{l} \frac{1}{2\pi \varepsilon_{0}}\cdot \ln\frac{r_{2}\left(x,y\right)r_{3}\left(x,y\right)}{r_{1}\left(x,y\right)r_{4}\left(x,y\right)}\;\mathrm{for}\;\mathrm{phase}\;2\;\mathrm{and}\;4\\ \frac{1}{2\pi \varepsilon_{0}}\cdot \ln\frac{1}{r_{1}\left(x,y\right)r_{2}\left(x,y\right)r_{3}\left(x,y\right)r_{4}\left(x,y\right)}\;\;\mathrm{for}\;\mathrm{phase}\;1\;\mathrm{and}\;3 \end{array}\right.$$
where
(16a)$$r_{1}\left(x,y\right)=\sqrt{{\left(x-x_{}^{\left(n\right)}\right)}^{2}+{\left(y-y_{}^{\left(n\right)}\right)}^{2}}$$
(16b)$$r_{2}\left(x,y\right)=\sqrt{{\left(x+x_{}^{\left(n\right)}\right)}^{2}+{\left(y-y_{}^{\left(n\right)}\right)}^{2}}$$
(16c)$$r_{3}\left(x,y\right)=\sqrt{{\left(x+x_{}^{\left(n\right)}\right)}^{2}+{\left(y+y_{}^{\left(n\right)}\right)}^{2}}$$
(16d)$$r_{4}\left(x,y\right)=\sqrt{{\left(x-x_{}^{\left(n\right)}\right)}^{2}+{\left(y+y_{}^{\left(n\right)}\right)}^{2}}$$

*n* represents the *n*-th fictitious charge. The electric field comes from the differentiation of the potential $\phi \left(x,y\right)$ with respect to *x* and *y*:(17)$$E_{x}\left(x,y\right)=-\frac{\partial P}{\partial x}Q$$
and
(18)$$E_{y}\left(x,y\right)=-\frac{\partial P}{\partial y}Q$$

### 2.3. Accuracy Evaluation

The modified standard norm error [26] is introduced for the evaluation of accuracy, which is defined by the sum of the deviation of potential at contour checkpoints, calculated from
(19)$$Error=\sqrt{{\sum_{i=1}^{m}\left[\frac{V-\phi_{i}\left(x_{i},y_{i}\right)}{V}\right]}^{2}/m}$$
where *ϕ_i_* (*x_i_*, *y_i_*) is the calculated potential at the *i*-th checkpoint, *m* is the total number of checkpoints, and *V* is the corresponding surface potential of the electrode (i.e., applied voltage on the electrode). This accuracy criterion is more rigorous than the maximum deviation previously used [23]. The electrode configuration parameters are as previously used by Kawamoto [27]. The width of the electrode is 300 µm, the pitch is 600 µm, and the amplitude of the voltage is 800 V. The following calculations are based on the same parameters.

The accuracy of the CSM is affected by the number and placement of fictitious charges and contour points. Normally, the collocated contour points are evenly spaced along the horizontal and vertical axes with pitches of *d*_1_ and *δ*_1_. The distance between the fictitious charge and the boundary, vertically and horizontally, are represented by *s*_1_ and *s*_2_, respectively:(20)$$s_{1}=k_{1}d_{1}$$
(21)$$s_{2}=k_{2}\delta_{1}$$

Here, *k*_1_ and *k*_2_ are the assignment factors deciding the position of fictitious charges. In Table 1, *n*_1_ and *n*_2_ represent *D*/*d*_1_ and *δ*/*δ*_1_, respectively, the numbers of contour points along the length and width of each electrode boundary. The total number of checkpoints is 250. Different combinations of *n*_1_, *n*_2_, *k*_1_, and *k*_2_ were tested to estimate the CSM numerical calculation accuracy. Phases 3 and 4 are symmetrical to phases 1 and 2 with different polarity, so the accumulated error are the same and only shown for phases 1 and 2 in Table 1. Evaluations and simulations were implemented on a personal computer with a 2.9 GHz processor and 16 GB RAM using Wolfram Mathematica. The calculation time for the CSM to obtain the error results is also presented.

The small accumulated errors confirm the accuracy of the CSM. By the appropriate arrangement of *n*_1_, *n*_2_, *k*_1_, and *k*_2_, the accumulated error can be decreased to as low as 0.02%. By keeping *k*_1_ and *k*_2_ constant, it is found that an increase in the simulation points can improve the calculation accuracy; however, further increases may impair this, as illustrated by the last three columns. This is because overly dense points may lead to an ill-conditioned matrix and a decrease in the solution accuracy. The test results with different combinations of *k*_1_ and *k*_2_ show that a reasonable arrangement of *k*_1_ and *k*_2_ can increase the accuracy of the CSM further.

## 3. Theory of the Boundary Element Method (BEM)

### 3.1. Formulation

In the CSM, it may be challenging to arrange the fictitious charges properly, which affects the accuracy of the results. In contrast, the BEM does not rely upon fictitious charges, and only boundary discretization is required. Figure 4 illustrates the basic parameters for deploying the BEM. *P* and *P_i_* are integration and observation points, respectively. $\Gamma_{l}^{*}$ is the electrode surface in the first quadrant of the coordinate system. *r*_1_, *r*_2_, *r*_3_, and *r*_4_ are the distances between the integration point and the observation points and image fictitious charges. Using Green’s second identity with the appropriate fundamental solution, a boundary integral equation can be found as follows [28]:(22)$$c_{i}u\left(P_{i}\right)=\int_{\Gamma }\frac{\partial u\left(P\right)}{\partial n}G\left(P,P_{i}\right)d\Gamma -\int_{\Gamma }\frac{\partial G\left(P,P_{i}\right)}{\partial n}u\left(P\right)d\Gamma$$
where *u*(*P*) is the solution of $\nabla^{2}u=0$ for P∈Ω. and
$$c_{i}=\left\{\begin{array}{l} 1\;\mathrm{for}\;P\mathrm{inside}\mathrm{the}\mathrm{region}\Omega \\ 1/2\;\mathrm{for}\;P\mathrm{on}\mathrm{the}\mathrm{smooth}\mathrm{boundary}\Gamma \\ 0\;\mathrm{for}\;P\mathrm{outside}u\mathrm{the}\mathrm{region}\Omega \end{array}\right.$$
and
(23)$$G\left(P,P_{i}\right)=\frac{1}{2\pi }\ln\frac{1}{r_{1}}$$

Equation (23) is the fundamental solution, where *r*_1_ is the distance between points *P* and *P_i_*. Denoting the boundary of the electrodes as Γ_1_, Γ_2_, …, Γ*_w_*, it can be deduced from Equation (22) that
(24)$$c_{i}u\left(P_{i}\right)=\sum_{l=1}^{w}\int_{\Gamma_{l}}\frac{\partial u\left(P\right)}{\partial n}G\left(P,P_{i}\right)d\Gamma -\sum_{l=1}^{w}\int_{\Gamma_{l}}\frac{\partial G\left(P,P_{i}\right)}{\partial n}u\left(P\right)d\Gamma$$

On account of the relation in [29],
(25)∫Γl∂GP,Pi∂ndΓ=1−ciδkl, forPi,P on the boundaryΓk,Γl,resp.
where $\delta_{kl}$ is the Kronecker delta, and noting that $u\left(P\right)=u_{l}$ (constant) on the boundary Γ*_l_*, it follows that
(26)$$\sum_{l=1}^{w}\int_{\Gamma_{l}}u\left(P\right)\frac{\partial G\left(P,P_{i}\right)}{\partial n}d\Gamma =\sum_{l=1}^{w}u_{l}\left(1-c_{i}\right)\delta_{kl}=\left(1-c_{i}\right)u_{k}$$

Using Equations (24) and (26), this can be transformed into
(27)$$u_{k}=\sum_{l=1}^{w}\int_{\Gamma_{l}}\frac{\partial u}{\partial n}G\left(P,P_{i}\right)d\Gamma$$

In addition, *G* in Equation (27) can be replaced with a modified fundamental solution *G_s_*, that is,
(28)GsP,Pi=12πlnr2r3r1r4     for phase 2 and 412πln1r1r2r3r4 for phase 1 and 3
where
(29a)$$r_{1}=\sqrt{{\left(x-x_{i}\right)}^{2}+{\left(y-y_{i}\right)}^{2}}$$
(29b)$$r_{2}=\sqrt{{\left(x+x_{i}\right)}^{2}+{\left(y-y_{i}\right)}^{2}}$$
(29c)$$r_{3}=\sqrt{{\left(x+x_{i}\right)}^{2}+{\left(y+y_{i}\right)}^{2}}$$
(29d)$$r_{4}=\sqrt{{\left(x-x_{i}\right)}^{2}+{\left(y+y_{i}\right)}^{2}}$$

Equation (29a–d) can be interpreted by the method of images, as shown in Figure 4. As a consequence, Equation (27) can be simplified to
(30)$$u_{k}=\sum_{l=1}^{w/2}\int_{\Gamma_{l}^{*}}\frac{\partial u}{\partial n}G_{s}\left(P,P_{i}\right)d\Gamma$$

Equation (30) reduces the unknowns, remarkably, to 25% of those in the original Equation (27).

The discretization of (30) yields
(31)$$u_{k}^{\left(i\right)}=\sum_{l=1}^{w/2}\sum_{j=1}^{N}\int_{\Gamma_{l,j}^{*}}\frac{\partial u\left(P\right)}{\partial n}G_{s}\left(P,P_{i}\right)d\Gamma ={\sum_{l=1}^{w}\sum_{j=1}^{N}\left(\frac{\partial u}{\partial n}\right)}_{l}^{\left(j\right)}G_{k,l}^{\left(i,j\right)}$$
where
(32)Gk,li,j=12π∫Γl,j∗lnr2r3r1r4dΓ    for phase 2 and 412π∫Γl,j∗ln1r1r2r3r4dΓ   for phase 1 and 3

${\left(\frac{\partial u}{\partial n}\right)}_{l}^{\left(j\right)}$ is assumed to be constant in each boundary element and needs to be solved. 

Accordingly, a linear equation system can be established with Equations (30) and (32):(33)$$\left[\begin{array}{ccccc} \begin{array}{l} G_{11}\\ G_{21} \end{array} & \begin{array}{l} G_{12}\\ G_{22} \end{array} & \begin{array}{l} .\\ . \end{array} & \begin{array}{l} .\\ . \end{array} & \begin{array}{l} G_{1,w/2}\\ G_{2,w/2} \end{array}\\ . & . & . & . & .\\ . & . & . & . & .\\ . & . & . & . & .\\ G_{w/2,1} & G_{w/2,2} & . & . & G_{w/2,w/2} \end{array}\right]\left[\begin{array}{l} u_{n}^{\left(1\right)}\\ u_{n}^{\left(2\right)}\\ \;.\\ \;.\\ \;.\\ u_{n}^{\left(w/2\right)} \end{array}\right]=\left[\begin{array}{l} s_{1}\\ s_{2}\\ \;.\\ \;.\\ \;.\\ s_{w/2} \end{array}\right]$$
where
(34)$$G_{kl}=\left[\begin{array}{ccccc} \begin{array}{l} G_{k,l}^{\left(1,1\right)}\\ G_{k,l}^{\left(2,1\right)} \end{array} & \begin{array}{l} G_{k,l}^{\left(1,2\right)}\\ G_{k,l}^{\left(2,2\right)} \end{array} & \begin{array}{l} .\\ . \end{array} & \begin{array}{l} .\\ . \end{array} & \begin{array}{l} G_{k,l}^{\left(1,N\right)}\\ G_{k,l}^{\left(2,N\right)} \end{array}\\ . & . & . & . & .\\ . & . & . & . & .\\ . & . & . & . & .\\ G_{k,l}^{\left(N,1\right)} & G_{k,l}^{\left(N,2\right)} & . & . & G_{k,l}^{\left(N,N\right)} \end{array}\right]$$
(35)$$u_{n}^{\left(l\right)}={\left[\begin{array}{ccc} \begin{array}{ccc} \begin{array}{cc} {\left(\frac{\partial u}{\partial n}\right)}_{l}^{\left(1\right)} & {\left(\frac{\partial u}{\partial n}\right)}_{l}^{\left(2\right)} \end{array} & . & . \end{array} & . & {\left(\frac{\partial u}{\partial n}\right)}_{l}^{\left(N\right)} \end{array}\right]}^{T}$$
(36)$$s_{l}={\left[\begin{array}{ccc} \begin{array}{ccc} \begin{array}{cc} u_{l} & u_{l} \end{array} & . & . \end{array} & . & u_{l} \end{array}\right]}^{T}$$

Equation (33) can be written more concisely as follows:(37)$$GU_{n}=S$$
where **U***_n_* is the solution vector representing the normal derivative of the potential on the electrode surfaces.

Using the obtained normal derivative ∂u/∂n on the electrode surfaces, the potential of *P* in the region Ω can be found by Equation (24) with *c_i_* = 1:(38)$$u\left(P_{i}\right)=\sum_{l=1}^{w}\int_{\Gamma_{l}}\frac{\partial u\left(P\right)}{\partial n}G\left(P,P_{i}\right)d\Gamma -\sum_{l=1}^{w}\int_{\Gamma_{l}}\frac{\partial G\left(P,P_{i}\right)}{\partial n}u\left(P\right)d\Gamma$$

The second term on the RHS of (38) vanishes due to $u\left(P\right)=u_{l}$ on Γ*_l_* and
(39)∫Γl∂GP,Pi∂ndΓ=0, for Pi inside the region Ω and P on the boundary Γl

Consequently, Equation (38) can be reduced to
(40)$$u\left(P_{i}\right)=\sum_{l=1}^{w}\int_{\Gamma_{l}}\frac{\partial u\left(P\right)}{\partial n}G\left(P,P_{i}\right)d\Gamma$$

Equation (40) can be simplified further by replacing *G* with *G_s_* as follows:(41)$$u\left(P_{i}\right)=\sum_{l=1}^{w/2}\int_{\Gamma_{l}^{*}}\frac{\partial u\left(P\right)}{\partial n}G_{s}\left(P,P_{i}\right)d\Gamma$$

Discretization of Equation (41) gives the following:(42)$$u\left(P_{i}\right)=G_{K}U_{n}$$
where
(43)$$\begin{array}{l} G_{K}=\left[G_{1},G_{2},\ldots ,G_{w/2}\right],G_{l}=\left[G_{l}^{\left(1\right)},G_{l}^{\left(2\right)},\ldots ,G_{l}^{\left(N\right)}\right]\\ \;l=1,2,\ldots ,w/2 \end{array}$$
with
(44)Glj=12π∫Γl,j∗lnr2r3r1r4dΓ,    for Phase 2 and 412π∫Γl,j∗ln1r1r2r3r4dΓ,  for Phase 1 and 3 j=1,2,…,N
and
(45)$$U_{n}={\left[u_{n}^{\left(1\right)},u_{n}^{\left(2\right)},\ldots ,u_{n}^{\left(w/2\right)}\right]}^{T}$$

Therefore, the *x*, *y* components of ***E*** can be calculated as follows:(46a)$$E_{x}\left(P_{i}\right)=-\frac{\partial G_{K}}{\partial x_{i}}U_{n}$$
and
(46b)$$E_{y}\left(P_{i}\right)=-\frac{\partial G_{K}}{\partial y_{i}}U_{n}$$

The computation efficiency can be improved further by solving the integrals (32) analytically with the approach given in [30]. However, more convenient results can be obtained for the evaluation of the integrals. An illustration of the integral model is shown in Figure 5. Parameter *s* is the arclength of a linear segment, and the integral point *P* (*x*, y) moves along the linear segment. (*x*_0_, *y*_0_) is the coordinate of the segment midpoint, and (*x_i_*, *y_i_*) is the coordinate of the observation point. 

The position of *P* (*x*, y) can be denoted by
(47)$$x=x_{0}+\beta s,\;y=y_{0}+\delta s$$
where *β* = *n_y_*, *δ* = −*n_x_*. It is worth mentioning that the normal vector always points to the left as *s* increases, and $\beta^{2}+\delta^{2}=1$. The integral of (32) can be expanded as the sum form of four integrals related to *r*_1_, *r*_2_, *r*_3,_ and *r*_4_, respectively. By this notation, each of the integrals relevant to (32) can be written as
(48)$$\frac{1}{2\pi }\int_{\Gamma }\ln\frac{1}{\left|P_{i}-P\right|}d\Gamma =\frac{1}{2\pi }\int_{-\gamma /2}^{\gamma /2}\ln\frac{1}{\sqrt{{\left(x_{0}+\beta s-x_{i}\right)}^{2}+{\left(y_{0}+\delta s-y_{i}\right)}^{2}}}ds$$
where *γ* is the length of the element. Moreover, by the substitution
(49)$$s=t-\kappa ,\kappa =\beta \left(x_{0}-x_{i}\right)+\delta \left(y_{0}-y_{i}\right)$$

The integral of (48) is equivalent to
(50)$$\begin{array}{l} \frac{1}{2\pi }\int_{-\gamma /2}^{\gamma /2}\ln\frac{1}{\sqrt{{\left(x_{0}+\beta s-x_{i}\right)}^{2}+{\left(y_{0}+\delta s-y_{i}\right)}^{2}}}ds=-\frac{1}{4\pi }\int_{\kappa -\gamma /2}^{\kappa +\gamma /2}\ln\left(t^{2}+u^{2}\right)dt\\ =-\frac{1}{4\pi }{\left[t\ln\left(t^{2}+u^{2}\right)-2t+2u\arctan\left(t/u\right)\right]}_{t=\kappa -\gamma /2}^{t=\kappa +\gamma /2}\\ =-\frac{1}{4\pi }\left\{\begin{array}{l} \kappa \ln\frac{{\left(\kappa +\gamma /2\right)}^{2}+u^{2}}{{\left(\kappa -\gamma /2\right)}^{2}+u^{2}}+\frac{\gamma }{2}\ln\left\{\left[{\left(\kappa +\gamma /2\right)}^{2}+u^{2}\right]\left[{\left(\kappa -\gamma /2\right)}^{2}+u^{2}\right]\right\}\\ -2\gamma +2u\left(\arctan\frac{\kappa +\gamma /2}{u}-\arctan\frac{\kappa -\gamma /2}{u}\right) \end{array}\right\} \end{array}$$
where
(51)$$u=\delta \left(x_{0}-x_{i}\right)-\beta \left(y_{0}-y_{i}\right)$$

Equation (50) is valid for u≠0. When *u* = 0, the following result can be employed:(52)$$\begin{array}{l} \frac{1}{2\pi }\int_{-\gamma /2}^{\gamma /2}\ln\frac{1}{\sqrt{{\left(x_{0}+\beta s-x_{i}\right)}^{2}+{\left(y_{0}+\delta s-y_{i}\right)}^{2}}}ds=-\frac{1}{2\pi }{\left[t\ln\left|t\right|-t\right]}_{t=\kappa -\gamma /2}^{t=\kappa +\gamma /2}\\ =-\frac{1}{2\pi }\left[\kappa \ln\left|\frac{\kappa +\gamma /2}{\kappa -\gamma /2}\right|+\frac{\gamma }{2}\ln\left|\kappa^{2}-\gamma^{2}/4\right|-\gamma \right] \end{array}$$

When *P*_0_(*x*_0_, *y*_0_) and *P_i_*(*x_i_*, *y_i_*) coincide, we have *κ* = *u* = 0, and the corresponding singular integral can be obtained easily using Equation (52):(53)$$\frac{1}{2\pi }\int_{-\gamma /2}^{\gamma /2}\ln\frac{1}{\sqrt{{\left(x_{0}+\beta s-x_{i}\right)}^{2}+{\left(y_{0}+\delta s-y_{i}\right)}^{2}}}ds=\frac{\gamma }{2\pi }\left(1-\ln\frac{\gamma }{2}\right)$$

Thus, the integrals of (32) can be found by proper linear combinations of the analytical solutions (50), (52), or (53). By this approach, no numerical quadrature is required, and the BEM algorithm can be implemented with very high efficiency.

### 3.2. Comparison of CSM, BEM, and FEM

The two above-mentioned approaches were used to calculate the electric field. Figure 6 and Figure 7 show the electric field at the height of 50 μm above an electrode surface with 8 and 16 electrodes, respectively. Clearly, the BEM and CSM produce highly consistent results, which confirms their validity. The code for the CSM can be found from the Appendix A.

In order to quantify the comparison for the accuracy of the CSM and BEM, the same process for calculating the error for the BEM was applied, and the results are shown in Table 2. *n*_1_ and *n*_2_ are the numbers on each long side and short side of boundary elements. It is clear that the CSM has slightly higher accuracy than the BEM. The CSM only needs 4.5 s to achieve the accuracy of 0.12% at phase, while the BEM needs 12 s. However, the CSM needs time to set up the positions of fictitious charges manually, which might be time-consuming for complex boundary conditions. The BEM does not need pre-processing, and the image method reduces unknown elements to 25% of those in the original equations, which greatly improves the computation efficiency of the BEM.

The finite element method (FEM) is frequently used to solve electrostatic problems [31,32]. The accuracy of the FEM is related closely to the quality of the calculating mesh used; at the edge of the conductors, extremely fine meshing is required, while further away from the electrodes, it can be coarser [33]. The area in the eight-electrode model was divided into two areas (blue and grey) by the rectangular boundary around the electrodes, as shown in Figure 8. A triangular mesh was applied. The element size outside the rectangular area is set as extremely fine, while the mesh size is set to be smaller than 4 × 10^−6^ m inside the rectangular area.

Comparisons between the electric field obtained by the CSM, BEM, and FEM for 8 and 16 electrodes are shown in Figure 9 and Figure 10. All methods are in good agreement in the central electrode region. However, the FEM results show significant deviation from the CSM and BEM on both sides, especially at the corners of the electrodes. It is clear that the CSM and BEM have higher accuracy than the FEM in solving this BVP. In addition, mesh generating is time-consuming with the FEM, and the impact of the increase in electrode number is more severe.

In addition, the FEM has a large deviation from the CSM and BEM in the area of the shear distribution of the electric field.

## 4. Electric Field and Dielectrophoretic Component Analysis

### 4.1. Distribution of Potential and Electric Field

The distribution of potential and electric field direction above eight electrodes in phases 1 and 2 are shown as contour lines and arrows in Figure 11a,b, from the BEM. The black bars on the bottom of the two figures signify each of the eight electrodes. The field line directions can be used to predict the particle motion when the Coulomb force dominates.

The electric field components *E_x_* and *E_y_* of eight electrodes in phase 2 are calculated and shown in Figure 12 and Figure 13, which accurately include the edge effect on the end of electrode arrays. For the field of a large number of parallel electrodes, the periodic symmetry of the field allows us to extend the solution with the same phase relationship by simply repeating the periodic solution (in the middle range of the calculated field) in the positive and negative x-directions.

### 4.2. Electric Fields with Different Electrode Thickness

Figure 14 compares the electric field magnitude with electrode thicknesses of 18 μm and 180 μm while maintaining the other model parameters the same. The electric field is evaluated at heights of 50 μm, 500 μm, and 1mm above the surface of the conveyor. At the height of 50 μm, the maximum value of the electric fields is similar. However, Figure 14b shows that electrodes with larger thickness have higher electric fields and that the difference is more distinct at higher altitudes. The effect of electrode thickness on the electric field is instructive for the design of ETW system electrode configurations.

### 4.3. Dielectrophoretic Component Analysis

The dielectrophoretic force has received increasing attention in particles or biological cell separation [34] and carbon nanotube manipulation [35]. The DEP force acts on particles with a dipole moment in a non-uniform electric field. The time-averaged force on the particle can be calculated as follows [17]:(54)Fdep=14vReα∇E˜⋅E˜∗−12vReα∇×E˜×E˜∗
where $\tilde{E}$ is a general complex amplitude of the electric field, * indicates a complex conjugate, *v* is the volume of the particle, and *α* is the effective polarizability related to particle and dielectric permittivity. In our case, the electric field is constant and calculated independently in each phase, so there is no phase variation, and the second term on the right side of Equation (54) is zero. Figure 15 shows the DEP potential $\left(\tilde{E}\cdot {\tilde{E}}^{*}\right)$ as contour lines and vector direction <*F*_dep_> as arrows.

### 4.4. Real Case of Using Dielectrophoresis

For the manipulation and transport of particles or biological cells, the theoretical analysis of different forces is crucial for the design and optimization of the conveyor system. For example, the charged particle can have both a Coulomb force and a dielectrophoretic force, and the two different forces may drive the particle to move in different directions. Figure 16 compares DEP and Coulomb forces acting on ballotini particles of various sizes and a charge of 10% of the saturation charge in the air. The saturation of particles is calculated by the relation of Pauthenier [36].
(55)$$q_{m}=4\pi r^{2}\varepsilon_{0}\frac{3\varepsilon_{p}}{\varepsilon_{p}+2}E_{c}$$
where *E_c_* is the dielectric strength of air and $E_{c}\approx 3\times 10^{6}\;V/m$; *r* is the particle size and *ε_p_* is the relative dielectric permittivity of the particle. The real Clausius–Mossotti factor for the particle is assumed as 0.5. The position of the particle is fixed at the middle of the first electrode and half of the width of the electrode above the surface of the electrode, which has the largest DEP force in the *y* direction. Because the DEP force is volume related, it becomes more dominant as the particle size increases. The comparison is useful for the design of a system to decide the dominance force on the particle. Further analysis could be obtained, for example, on the trajectory simulation by the action of the two forces. 

## 5. Conclusions

The CSM and BEM were explored for the numerical solution of the electric field for an interdigitated, rectangular electrode array with a specified thickness. These two methods are readily implemented with the method of images and can achieve high computational efficiency and accuracy. In addition, analytical solutions for the integral of Green’s function along the boundary elements are derived. These analytical formulas are beneficial to the efficient implementation of the BEM and can be utilized for the general 2D BEM of electrostatic problems.

The boundary conditions used in the CSM and BEM numerical calculation do not require simplification. Electric field and DEP component analysis with 8- and 16-electrode systems were provided, which showed accurate results near the electrode surface. These results can be used for the design of different ETW electrode configurations for particle transport and biological cell manipulations. The FEM results were compared with those of the CSM and BEM and showed differences at the electrode ends, indicating that the FEM may not be suitable for determining the electric field in systems with sharp boundaries. Moreover, the CSM and BEM are less time-consuming. Overall, the CSM and BEM are more general numerical methods for dealing with electrostatic field problems and can be adapted readily to more complex boundary conditions, such as a 3D model of the electrode array.

This accurate evaluation of the electric field could potentially benefit the analysis of particles and cells in transport and separation, as the estimation of particle trajectory is highly sensitive to the electric field. 

## Figures and Tables

**Figure 1 micromachines-14-01347-f001:**
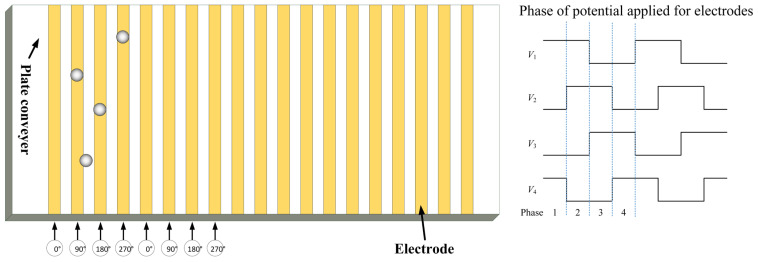
Diagram showing the typical application system and applied voltage.

**Figure 2 micromachines-14-01347-f002:**
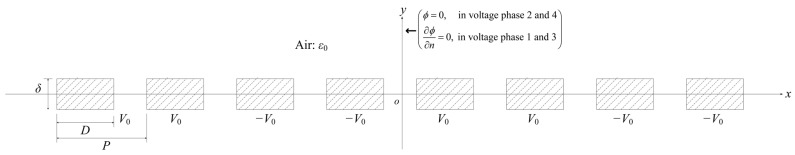
Diagram showing the typical application system, consisting of the interdigitated electrode array. D is the electrode width, p is the electrode pitch, and δ is the electrode thickness. The voltage on the boundaries is the example at phase 2.

**Figure 3 micromachines-14-01347-f003:**
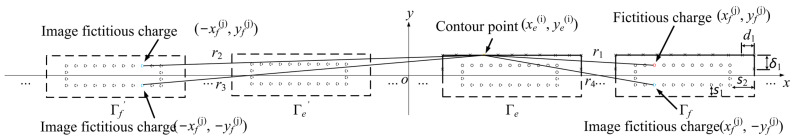
Allocation of contour points and fictitious charges with the method of images.

**Figure 4 micromachines-14-01347-f004:**
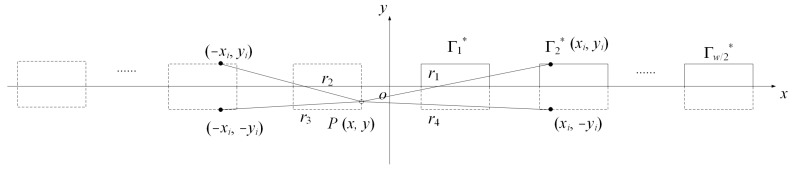
Electrode geometry for the BEM and the method of images.

**Figure 5 micromachines-14-01347-f005:**
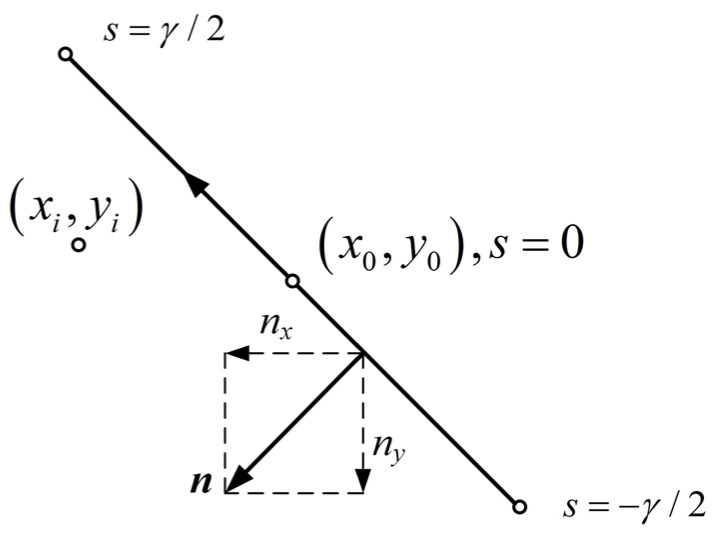
Arclength parameters for the element integration.

**Figure 6 micromachines-14-01347-f006:**
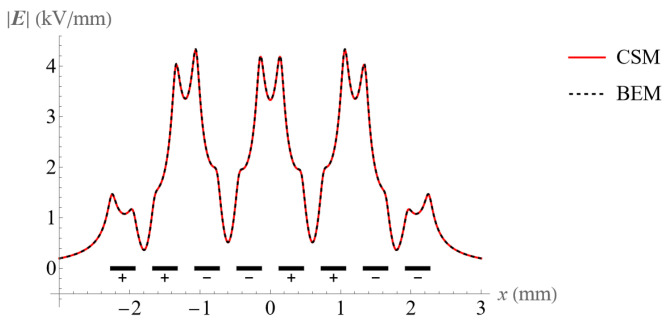
Comparison of the electric field at the height of 50 μm above the 8 electrodes using the BEM and CSM.

**Figure 7 micromachines-14-01347-f007:**
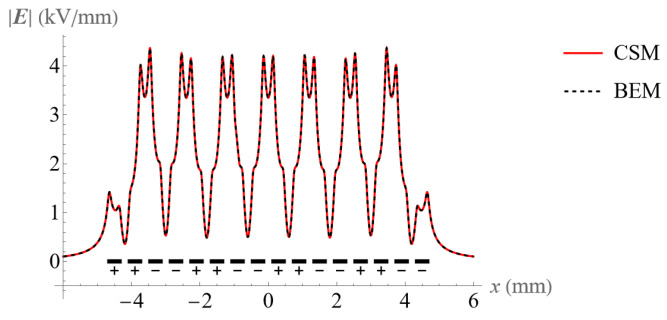
Comparison of the electric field at the height of 50 μm above the 16 electrodes using the BEM and CSM.

**Figure 8 micromachines-14-01347-f008:**
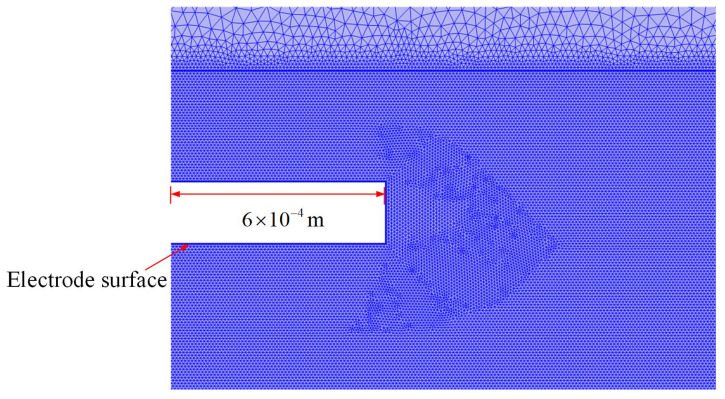
Subdivision area illustration.

**Figure 9 micromachines-14-01347-f009:**
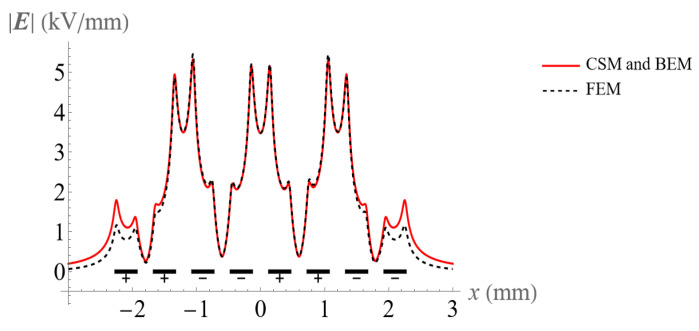
Comparison of the electric field at the height of 32 μm above the 8 electrodes using the CSM, BEM, and FEM.

**Figure 10 micromachines-14-01347-f010:**
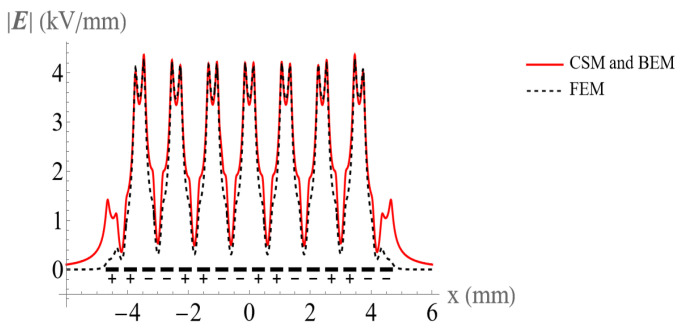
Comparison of the electric field at the height of 50 μm above the 16 electrodes using the CSM, BEM, and FEM.

**Figure 11 micromachines-14-01347-f011:**
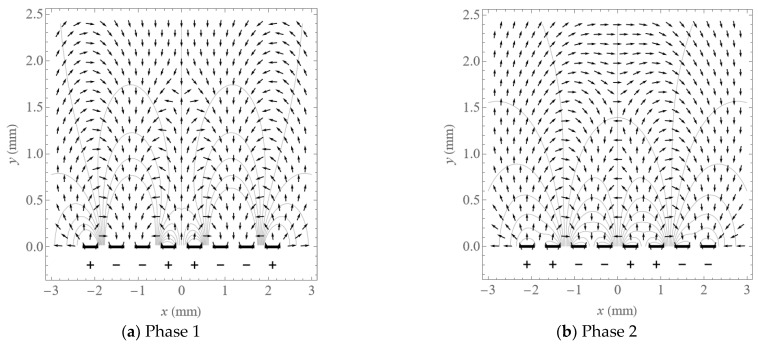
Contour plot of potential and vector plot of electric field above electrodes.

**Figure 12 micromachines-14-01347-f012:**
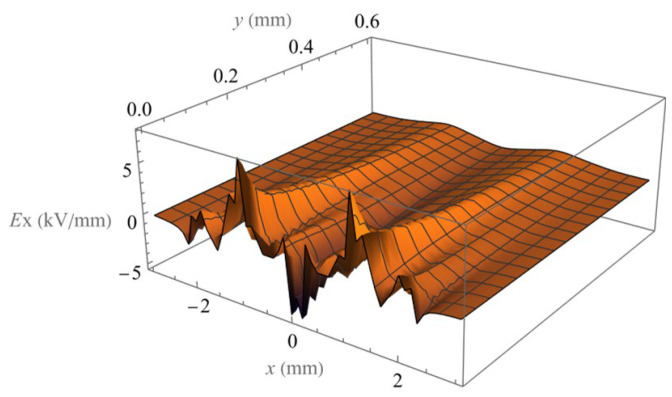
Spatial distribution of the magnitude of field component Ex.

**Figure 13 micromachines-14-01347-f013:**
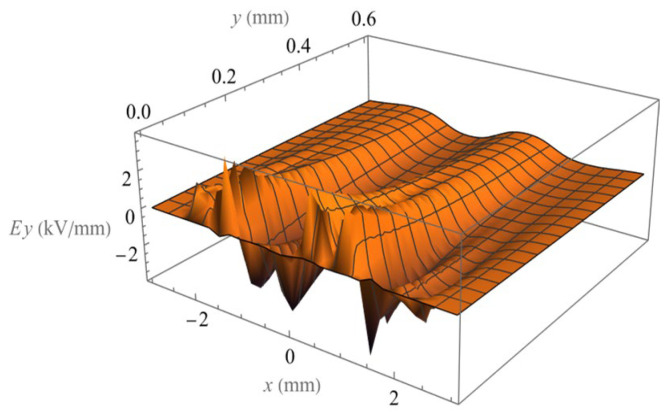
Spatial distribution of the magnitude of field component Ey.

**Figure 14 micromachines-14-01347-f014:**
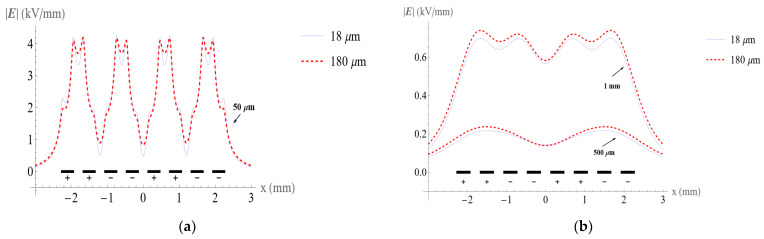
Comparison of the electric field distribution with 18 μm and 180 μm electrode thicknesses (**a**) at the height of 50 μm above the surface of the conveyor and (**b**) at the height of 500 μm and 1 mm above the surface of the conveyor.

**Figure 15 micromachines-14-01347-f015:**
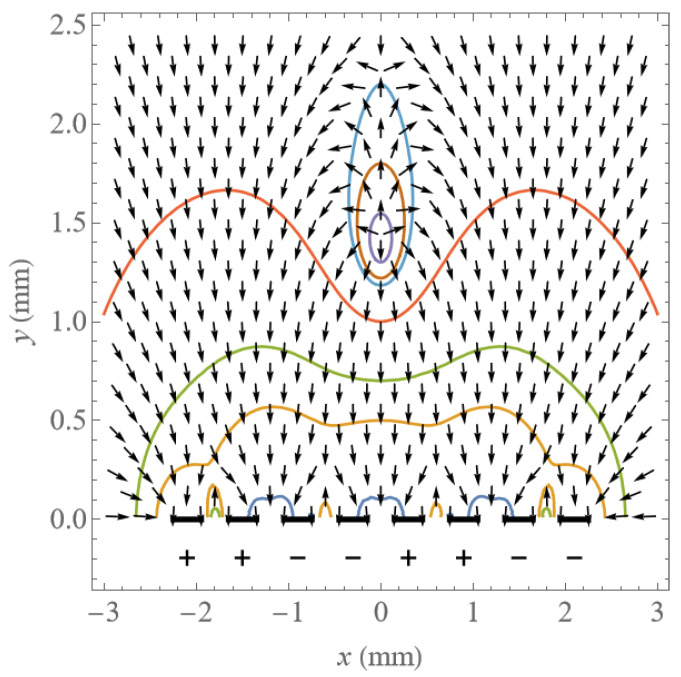
Contour plot of DEP potential and vector plot of DEP above electrodes in phase 2.

**Figure 16 micromachines-14-01347-f016:**
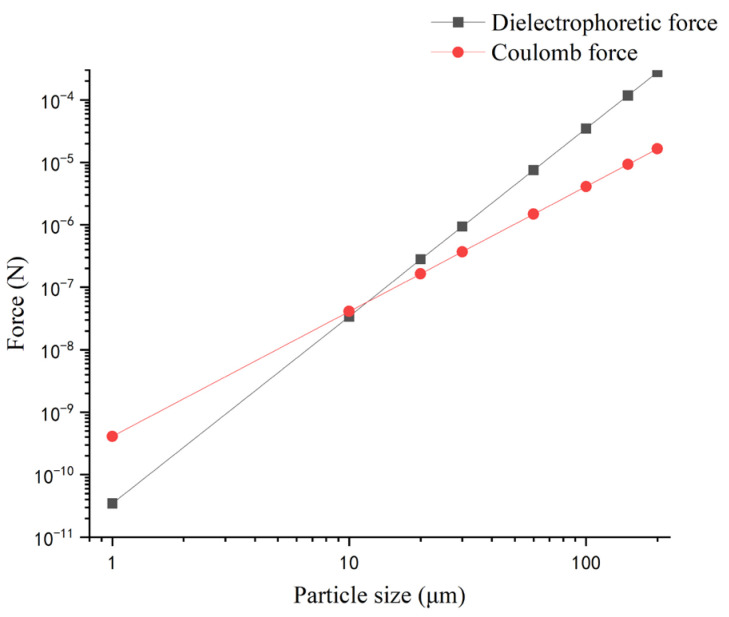
Comparison between DEP and Coulomb force.

**Table 1 micromachines-14-01347-t001:** Standard norm error on each electrode in two different phases with a varying number of calculating points for the CSM.

Calculating parameters	*k*_1_ = 1/6 *k*_2_ = 1/6	*n*_1_ = 200 *n*_2_ = 10	*n*_1_ = 150 *n*_2_ = 10	*n*_1_ = 100 *n*_2_ = 10	*n*_1_ = 100 *n*_2_ = 15	*n*_1_ = 100 *n*_2_ = 20
Standard norm error	Phase 1	0.03%	0.04%	0.07%	0.08%	0.12%
	Phase 2	0.02%	0.03%	0.05%	0.03%	0.09%
Time		11.6 s	7.6 s	3.9 s	4.1 s	4.5 s
Calculating parameters	*n*_1_ = 200 *n*_2_ = 10	*k*_1_ = 1/6 *k*_2_ = 1/6	*k*_1_ = 1/5 *k*_2_ = 1/5	*k*_1_ = 1/3 *k*_2_ = 1/3	*k*_1_ = 1/2 *k*_2_ = 1/2	*k*_1_ = 1/3 *k*_2_ = 1/6
Standard norm error	Phase 1	0.03%	0.03%	0.04%	0.06%	0.06%
	Phase 2	0.02%	0.02%	0.03%	0.05%	0.04%
Time:		12.0 s	12.1 s	12.3 s	12.1 s	12.1 s

**Table 2 micromachines-14-01347-t002:** Standard norm error on each electrode in two different phases with a varying number of calculating points for the BEM.

Calculating parameters		*n*_1_ = 200 *n*_2_ = 10	*n*_1_ = 150 *n*_2_ = 10	*n*_1_ = 100 *n*_2_ = 10	*n*_1_ = 100 *n*_2_ = 15	*n*_1_ = 100 *n*_2_ = 20
Accumulated error	Phase 1	0.13%	0.16%	0.19%	0.19%	0.18%
	Phase 2	0.10%	0.12%	0.15%	0.14%	0.13%
Time		12.0 s	8.1 s	4.9 s	5.3 s	5.4 s

## Data Availability

The data from this work will be shared and made available upon request to the corresponding author.

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
