# Peer review of "Numerical Solution of the Electric Field and Dielectrophoresis Force of Electrostatic Traveling Wave System"

_micromachines, 2023, doi:10.3390/mi14071347_

Round 1
Reviewer 1 Report
Thanks to the author for bringing this modeling method to describe the electric field and dielectrophoresis force in the electrostatic traveling waver system
I have a few questions regarding the materials discussed
1. Figure 3 and 4 These three figure believed to be the primary figure to explain the application system and how it been established by math language. However I did not saw enough explanation regarding these three graphs. Please define all the parameters more clearly.
2. Line 320—how to define computation efficiency here? If possible you can list the pros and cons of all three methods here?
3. Section 4.3. Is that possible you can give few examples of real case applications here?
The explanation of figure notes need to be improved
Author Response
We are deeply grateful for the valuable comments from Reviewer 1. We have read through comments carefully and have made corrections.
The Point-by-Point Responses have been completed. Please see the attachment. All of our answers are highlighted with red and blue color in the following contents. .

Reviewer 2 Report
See attached report.

The issues with English language use in the paper are predominantly related to missing subjects (usually "the" is missing before BEM or CSM) or punctuation (mostly on equations). Below is a short list of other minor issues:
Line 73: "both two" -> "both of the following two"
Caption to Figure 4: "and methods of images." -> "with the method of images."
Line 270: missing subject: "Illustration of..." -> "An illustration of..."
Line 302 contains a fragment of a sentence with no full stop- I think it should be "The finite element method (FEM) is frequently used to...."
Author Response
We would like to express our grateful thanks to Reviewer 2 for the careful and patient revision, as well as helpful and critical comments on the manuscript.
The Point-by-Point response has been completed. Please see the attachment. All of our answers are highlighted in red and blue.

Round 2
Reviewer 2 Report
The authors have adequately addressed my original list of comments and I am happy to recommend publication of the manuscript in Micromachines.
There are a couple of the points that could still do with some work though, but these are minor enough that I do not think they require further review and can be dealt with at the editorial stage if necessary:
1. I would still recommend to be more precise in the abstract about what the word "properly" means in the context of "properly arranging the positions and numbers of contour points and fictitious charges". What would constitute improperly arranging them? Do they need to be kept away from the electrode boundaries? Is there a rule of thumb for "properly arranging"? It is the word "properly" that is vague and non-scientific since it carries no specific meaning or information!
2. Figure 3 is now clearer but does not appear to fit properly into the space provided now! The text at the bottom has been partially truncated.... an easy fix I'm sure.
The authors have adequately addressed my original list of comments - thanks!